# The Effect of Priority Access of Dentists to COVID-19 Vaccination in the Czech Republic

**DOI:** 10.3390/vaccines10081252

**Published:** 2022-08-04

**Authors:** Vojtech Perina, Jan Schmidt

**Affiliations:** 1Department of Oral and Maxillofacial Surgery, Faculty of Medicine, Masaryk University and University Hospital Brno, 625 00 Brno, Czech Republic; 2Department of Dentistry, Faculty of Medicine in Hradec Kralove, Charles University and University Hospital Hradec Kralove, 500 05 Hradec Kralove, Czech Republic

**Keywords:** COVID-19, prevalence, vaccination, priority, dentist

## Abstract

The lack of vaccines in the first half of 2021 led to the need to prioritize access to vaccination. This approach has been associated with a number of issues, including ethics and effectiveness. However, analyses providing data on this topic are scarce. This work describes the effect of a priority approach to vaccination on the different development of the pandemic between Czech dentists and the Czech general population. The dentist-related data were obtained from survey studies published in 2021 and 2022, and the Czech general population data were mined from the Our World in Data online database. The analysis shows that until the beginning of vaccination, i.e., in December 2020, the prevalence of laboratory-confirmed COVID-19 infection among dentists was higher than in the general population by 22.5% (8.65% vs. 6.70%). This trend was reversed already in the first month after the start of vaccination, and the difference increased every month. Finally, in June 2021, priority vaccination statistically significantly (*p* < 0.0001) reduced the resulting prevalence among dentists (12.67%) compared to the general population (15.55%), which is a difference of 18.5%. This represents a prevalence shift between the populations by 40% during 6 months of priority vaccination. The results support the conclusion that the priority vaccination of healthcare workers was not only ethical but also rational and effective.

## 1. Introduction

As COVID-19 vaccine supplies were limited at the start of 2021, governments were forced to consider how to distribute them effectively. Many countries across Europe have decided to give healthcare professionals priority access to vaccination at the cost of delaying vaccination of the general population. This led to a parallel existence of two groups within the same society: one with a high risk of COVID-19 infection but highly vaccinated, the other with a normal risk of infection but without access to vaccination. This approach has been associated with a number of issues, including ethics and effectiveness [1]. Although ethical views may differ based on various philosophical approaches, the effectiveness of this model can only be evaluated based on objective data. However, despite the great interest of the scientific community in COVID-19 and the general support for prioritizing the vaccination of healthcare workers, studies assessing the resulting effect of vaccination on the COVID-19 prevalence among healthcare professionals are scarce, and more studies are needed [2,3].

Healthcare workers are at a higher risk of COVID-19 infection while working with people particularly vulnerable to the disease [3,4]. One of the healthcare occupational groups most at risk of COVID-19 infection are dentists. Their work is associated with the formation of a large amount of aerosol and, therefore, with a high risk of transmission of droplet infection [5]. Additionally, their work is performed in close proximity to patients, which further increases the infection risk. As such, they may serve as a suitable subpopulation of healthcare workers to demonstrate the effect of priority vaccination. Czech dentists are one of the best epidemiologically described professional groups during the COVID-19 pandemic and, so far, have shown a very high willingness to be vaccinated [6,7]. Additionally, Czech dentistry remained highly operative even during the pandemic [8]. In 2021, the Czech Republic was one of the countries with a high emphasis placed on the COVID-19 vaccination of healthcare professionals, even at the cost of 6 months restriction on the general population’s access to vaccination. At this time, the Czech Republic was the fourth most affected country by COVID-19 in the world [9,10]. This makes the Czech Republic and Czech dentists an ideal field for demonstrating the effect of priority vaccination. To the best of our knowledge, this is the first work on this topic.

The aim of this letter is to provide data on COVID-19 prevalence and vaccination in the population of Czech dentists during the first half of 2021 and its comparison to the Czech general population. Such data are important not only as an argument in the discussion on the appropriateness of prioritizing vaccination during the COVID-19 pandemic but may also contribute to strategic decisions in future pandemics [11].

## 2. Materials and Methods

### 2.1. Data Collection

The data in this letter are based on COVID-19 prevalence and COVID-19 vaccination data of Czech dentists and the Czech general population. The dentist-related data were obtained from studies performed and published by the authors in 2021 and 2022, and the Czech general population data were mined from the Our World in Data online database [6,7,12,13]. In both groups, only cases confirmed by PCR test were included. Study methods are described in detail in the publications mentioned above.

The time interval for the evaluation of prevalence was chosen from the beginning of the COVID-19 pandemic in the Czech Republic, i.e., March 2020, until June 2021. The time interval for the evaluation of COVID-19 vaccination was chosen from the beginning of vaccination in the Czech Republic, i.e., January 2021, until June 2021. June 2021 was chosen as the end of the reference period, as measures restricting the access of the Czech general population to COVID-19 vaccination ceased to be in force from this month.

### 2.2. Data Analysis

The data were organized and analyzed in Microsoft Office Excel (version 2106 for Windows, Microsoft Corporation, Redmond, WA, USA) and GraphPad Prism (version 8.0.0 for Windows, GraphPad Software, San Diego, CA, USA). Prevalence and vaccination data are presented as a percentage of the study participants and within the entire general Czech population. For statistical analysis, the chi-square test with Yates’ correction was used. Statistical significance was determined as *p* < 0.05 and labeled as follows: * *p* < 0.05, ** *p* < 0.001, *** *p* < 0.001, **** *p* < 0.0001. The Woolf logit confidence interval was used for the odds ratio (OR) and 95% confidence interval (CI) calculation.

## 3. Results

By 30 June 2021, a total of 1,667,287 individuals of the Czech general population were reported as positively PCR tested for COVID-19. The data were chronologically sorted and expressed as a percentage of the Czech general population. By the same date, out of the 2716 dentists involved in the study, representing 24.3% of all Chamber members, 344 stated they were positively tested for COVID-19 and provided the date of when the infection started. Data were chronologically sorted and expressed as a percentage of study participants. By January 2021, the prevalence was higher among dentists compared to that in the general population. However, since February 2021, this trend changed, and the prevalence of the general population exceeded the prevalence of dentists. From March to June 2021, this difference was statistically significant (March 2021: *p* = 0.0033, OR = 1.189, 95% CI 1.060 to 1.333; April 2021: *p* = 0.001, OR = 1.253, 95% CI 1.118 to 1.404; May 2021: *p* < 0.0001, OR = 1.268, 95% CI 1.132 to 1.420; June 2021: *p* < 0.0001, OR = 1.269, 95% CI 1.133 to 1.421). The results are presented in Figure 1. The relative difference in prevalence between Czech dentists and the general population was +22.5% at the end of December 2020 (8.65% vs. 6.70%) and −18.5% at the end of June 2021 (12.67% vs. 15.55%).

As Czech dentists were highly motivated to get vaccinated against COVID-19, the vaccination rate increased sharply every month. In contrast, vaccination among the general population increased very slowly as restrictions on access to vaccination limited it and only selected most-at-risk groups were vaccinated [14]. An increase in the vaccination rate of the general population in June 2021 is associated with an end of most of the restrictions during this month. By 30 June 2021, out of the 2716 dentists involved in the study, 2016 (79.53%) stated they were fully vaccinated against COVID-19. By the same time, a total of 3,308,807 (30.95%) individuals of the Czech general population were fully vaccinated against COVID-19. The difference in vaccination between dentists and the general population was statistically significant (January–June 2021: *p* < 0.0001). The results are presented in Figure 2.

The results demonstrate the priority vaccination’s effect on prevalence among dentists and, by extension, among healthcare professionals in general.

## 4. Limitations

As the analysis is not based on data obtained under fully controlled model conditions, its result may be affected by additional variables. For illustration, the Czech government granted an exception for particularly vulnerable population groups, such as the seriously ill and the very old. They were allowed to get vaccinated already during the spring of 2021. This fact could slightly reduce the difference between the prevalence among dentists and the general population. In a model situation in which only healthcare workers would be vaccinated, it can be expected that the effect of priority vaccination would be even more significant. Furthermore, the study population consists of individuals with a university education in the medical field, which may also have influenced the results. Among medically educated people, higher trust in evidence-based medicine, higher resistance to conspiracy theories, better access to protective equipment, or a higher willingness to get vaccinated can be expected [5,15,16].

## 5. Discussion and Conclusions

Data from January to June 2021 show that priority vaccination had a major impact on the lower COVID-19 prevalence among dentists compared to the general population in the Czech Republic. Until the beginning of vaccination, the prevalence among dentists was higher than in the general population. This trend was reversed already in the first month after the start of vaccination, and the difference increased every month. The gradual deceleration in prevalence among the general population since March 2021 can be associated with several factors, including the reduction of vector transmission through healthcare workers and the beginning of vaccination of the most-at-risk population over 70 years of age in March 2021.

Before vaccination, the prevalence among dentists was higher by about 2% in absolute terms and by 22.5% in relative terms. However, after vaccination, this trend was significantly reversed. In June 2021, priority vaccination led to a resulting prevalence of 12.67% among dentists and 15.55% among the general population representing a difference of 18.5%. To show the effect of vaccination, it is optimal to compare the relative difference between dentists and the general population before and after the vaccination. In December 2020, the prevalence among dentists was 22.5% higher compared to the general population, and in June 2021, 18.5% lower. This represents a prevalence shift between the populations by 40% during 6 months of priority vaccination. Such an outcome shows how the priority vaccination of healthcare workers notably reduced the spread of COVID-19 in this professional group. Additionally, by extension, the reduction in COVID-19 prevalence had a positive effect on the healthcare system’s operativeness and reduced the spread of infection within healthcare facilities, whose clients are usually the most vulnerable to COVID-19.

The results of this analysis support the conclusion that the priority vaccination of healthcare workers was not only ethical but also rational and effective.

## Figures and Tables

**Figure 1 vaccines-10-01252-f001:**
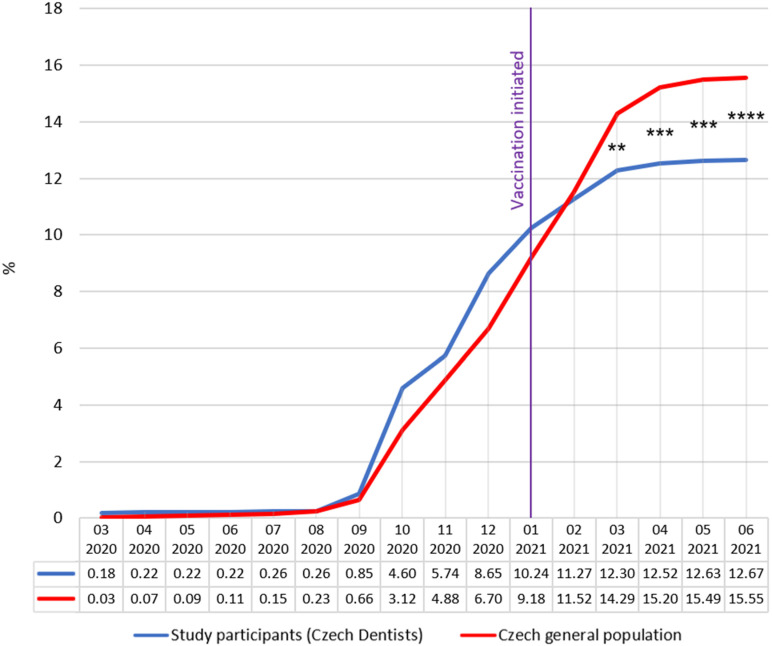
COVID-19 prevalence among Czech dentists and the Czech general population. The purple line indicates the time point when COVID-19 vaccination started. Chi-square test with Yates’ correction, ** *p* < 0.001, *** *p* < 0.001, **** *p* < 0.0001.

**Figure 2 vaccines-10-01252-f002:**
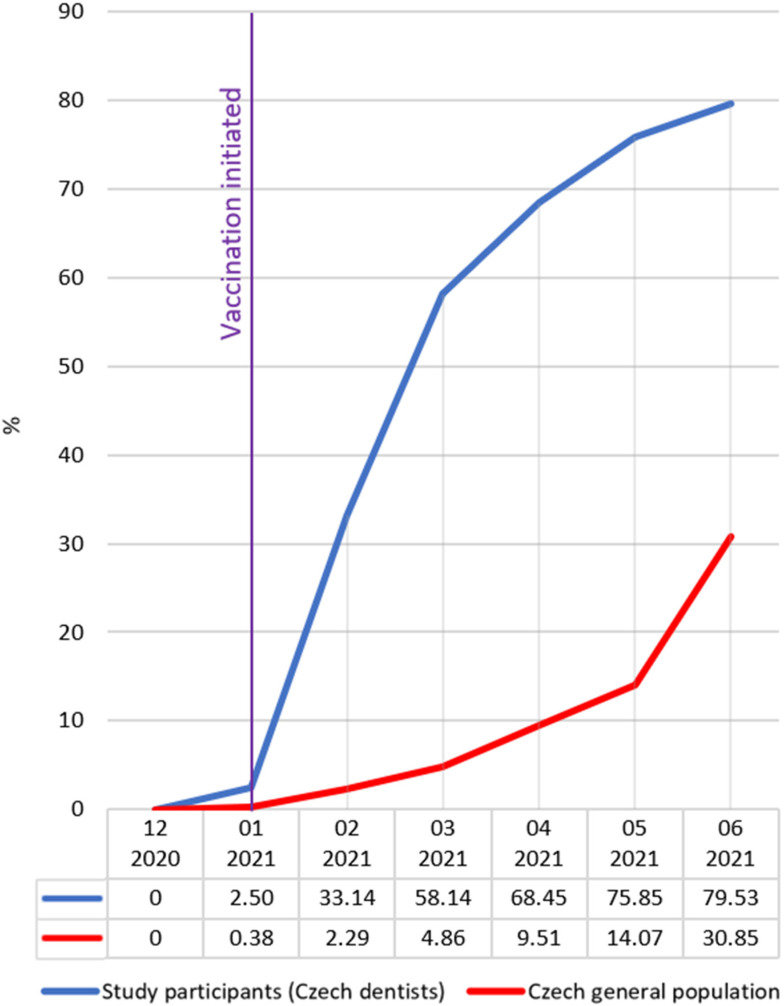
COVID-19 vaccination among Czech dentists and the Czech general population. The purple line indicates the time point when COVID-19 vaccination started.

## Data Availability

The dataset is available on demand from the corresponding author.

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
