# Peer review of "The Effect of Priority Access of Dentists to COVID-19 Vaccination in the Czech Republic"

_vaccines, 2022, doi:10.3390/vaccines10081252_

Round 1

Reviewer 1 Report

This is an observational study. It was aimed to study the effect of priority access of dentists to COVID 19 vaccination in the Czech Republic. Register data of PCR-confirmed COVID 19 infection before and six months after vaccination among dentists and general population were used. They found that the priority vaccination of healthcare workers significantly reduced the spread of COVID 19. The study design is proper, and the finding is important for the field and future vaccination strategy. I have several concerns which should be addressed first.

Major comments:

1. Although vaccination is one of the important factors for control and prevention of COVID 19 spread, many other factors e.g. non-pharmacological measures could also contribute to the reduced prevalence of COVID 19 infection in dentists, especially in early time e.g Feb 2021 when the vaccination was just started. This should be mentioned and discussed.

2. The prevalence of 12.67% among dentists and 15.55% among the general population representing a difference of 18.03%. How the percentage was obtained. It should be 18.5% not 18.03%?

3. Line17, the sentence "the prevalence among..." was not clear. should be changed to "the prevalence of laboratory confirmed COVID 19 infection among...". Similar change should be done in the text.    

Reviewer 2 Report

I read with great interest the article entitled ‘The effect of priority access of dentists to COVID-19 vaccination in the Czech Republic’

The work was aimed to describes the effect of a priority approach to vaccination on the different development of the pandemic between Czech dentists and the Czech general population.

Major concerns.

Although the subject of the study is interesting and well write, some aspects should be clarified.

a) The type of paper chosen is not explicitly described among those reported in "information for authors" provided by Vaccines. It is suggested to ask the Editorial Office for an opinion on the opportunity to submit the paper as a research article.

b) The work would be improved by the explanation of why dentists, especially at the beginning of the pandemic, were among the most affected categories. [i.e. Nioi, Matteo, et al. "COVID-19 and Italian healthcare workers from the initial sacrifice to the mRNA vaccine: pandemic chrono-history, epidemiological data, ethical dilemmas, and future challenges." Frontiers in Public Health 8 (2021): 591900.]. The reference is to the specific risk (drops of saliva in the air, other risky practices).

c)   There are many papers already advocating the need for a priority for health professionals. [Craxì, Lucia, et al. "Who should get COVID-19 vaccine first? A survey to evaluate hospital workers’ opinion." Vaccines 9.3 (2021): 189.; Thorsteinsdottir, Bjorg, and Bo Enemark Madsen. "Prioritizing health care workers and first responders for access to the COVID19 vaccine is not unethical, but both fair and effective–an ethical analysis." Scandinavian journal of trauma, resuscitation and emergency medicine 29.1 (2021): 1-3.; Haq, Mohsina, et al. "Identifying higher risk subgroups of health care workers for priority vaccination against COVID-19." Therapeutic advances in vaccines and immunotherapy 10 (2022): 25151355221080724.; Chirico, Francesco, Gabriella Nucera, and Nicola Magnavita. "COVID-19: protecting healthcare workers is a priority." Infection Control & Hospital Epidemiology 41.9 (2020): 1117-1117.]. What are the innovations that the current work introduces compared to the articles already published?

d) Authors are asked to analyze why vaccinations have been slow in the general population. [Nioi, Matteo, and Pietro Emanuele Napoli. "The Waiver of Patent Protections for COVID-19 Vaccines during the ongoing Pandemic and the Conspiracy Theories: Lights and Shadows of an Issue on the Ground." Frontiers in Medicine 8 (2021); Sallam, Malik. "COVID-19 vaccine hesitancy worldwide: a concise systematic review of vaccine acceptance rates." Vaccines 9.2 (2021): 160.]

e) Did the authors consider other possible variables that could explain the data? For example, a greater knowledge of the pathology or a more careful use of PPE or greater control over patients at the entrance? On what basis do they consider vaccines the only variable that can explain the difference? The theme should be expanded in the discussion.

f) The references must be expanded.

g) It is necessary that the authors introduce a chapter on the limitations of the study.

h) Considering the high number of articles on the subject, it is necessary that the authors give a satisfactory answer to each of the single points reported.

Round 2

Reviewer 2 Report

The authors improved the paper according to the suggestions given.